



# Technical Note: Low cost stage-camera system for continuous water level monitoring in ephemeral streams

Simone Noto[1,*], Flavia Tauro[2,*], Andrea Petroselli[3], Ciro Apollonio[4], Gianluca Botter[1], and Salvatore Grimaldi[2]

[1]Department of Civil, Environmental and Architectural Engineering, University of Padua, 35131 Padova, Italy
[2]Department for Innovation in Biological, Agro-food and Forest Systems, University of Tuscia, 01100 Viterbo, Italy
[3]Department of Economy, Engineering, Society and Business, University of Tuscia, 01100 Viterbo, Italy
[4]Department of Agriculture and Forest Sciences, University of Tuscia, 01100 Viterbo, Italy
[*]These authors contributed equally to this work.

**Correspondence:** Salvatore Grimaldi (salvatore.grimaldi@unitus.it)

**Abstract.** Monitoring ephemeral and intermittent streams is a major challenge in hydrology. While direct field observations are best to detect spatial patterns of flow persistence, on site inspections are time and labor intensive and may be impractical in difficult-to-access environments. Motivated by latest advancements of digital cameras and computer vision techniques, in this work, we describe the development and application of a stage-camera system to monitor the water level in ungauged headwater streams. The system encompasses a consumer grade wildlife camera with near infrared (NIR) night vision capabilities and a white pole that serves as reference object in the collected images. Time-lapse imagery is processed through a computationally inexpensive algorithm featuring image quantization and binarization, and water level time series are filtered through a simple statistical scheme. The feasibility of the approach is demonstrated through a set of benchmark experiments performed in controlled and natural settings, characterized by an increased level of complexity. Maximum mean absolute errors between stage-camera and reference data are approximately equal to 2 cm in the worst scenario that corresponds to severe hydrometeorological conditions. Our preliminary results are encouraging and support the scalability of the stage-camera in future implementations in a wide range of natural settings.

## 1 Introduction

Headwater streams govern runoff formation and are important controls for the ecological and geomorphic dynamics of catchments (Bishop et al., 2008; Datry et al., 2014; Siebers et al., 2020). Such upland channels exhibit a complex behavior dominated by the interplay of physical factors with pronounced spatial and temporal variability that result in network expansion and contraction cycles (Wigington Jr et al., 2005; Jaeger et al., 2007; Godsey and Kirchner, 2014; Goulsbra et al., 2014; Costigan et al., 2015; Peirce and Lindsay, 2015; Shaw, 2016; Whiting and Godsey, 2016; Shaw et al., 2017; Zimmer and McGlynn, 2017; Flo-





riancic et al., 2018; Jensen et al., 2018; Lovill et al., 2018; Ward et al., 2018; Barefoot et al., 2019; Garbin et al., 2019; Jaeger et al., 2019; Jensen et al., 2019; Botter and Durighetto, 2020; Durighetto et al., 2020). Seasonal and event-based variability of the climatic forcing in these uppermost areas often originate intermittent and ephemeral streams, whereby discontinuous flow or sporadic flow in response to rainfall events posit severe challenges to a thorough characterization of network patterns (Borg Galea et al., 2019; van Meerveld et al., 2020). Monitoring and mapping the onset and duration of flow in headwater streams is key to unravel the complex relationships between climatic and landscape features in temporary rivers.

Traditional observational approaches, including stream gauges and current meters, may often be inadequate to monitor streamflow in these dynamic and highly heterogeneous systems (Tauro et al., 2018). Overland flow detectors were developed by Kirkby et al. (1976) and modified versions have been applied in several field studies (Vertessy and Elsenbeer (1999); Vertessy et al. (2000); Elsenbeer and Vertessy (2000); Johnson et al. (2006); Zimmermann et al. (2014); Perez et al. (2020)). Such passive instrumentation can be deployed onto the soil surface and then manually checked to assess the presence or absence of water. Alternative methodologies have been conceived to automatically monitor the presence of water in ephemeral streams. For instance, temperature sensors have been used to sense temperature variations in the streambed following water occurrence (Constantz, 2008). Similarly, electrical resistance sensors have been proposed to sense the presence of water above the level of electrodes (Peirce and Lindsay, 2015). Modified temperature sensors through electrical conductivity (EC) have also been used to detect flow intermittency (Jensen et al., 2019; Paillex et al., 2020). High spatio-temporal resolution EC data has been achieved in (Assendelft and van Meerveld, 2019; Kaplan et al., 2019) through a multi-sensor system including a combination of electrical resistance, temperature, and flow sensors.

Data collected from temperature-based or EC systems can be affected by noise and are generally difficult to interpret. Notably, ambiguities in detecting zero-flow readings have been shown to have far-reaching implications for hydrological and biogeochemical predictions (Leigh et al., 2016; Vander Vorste et al., 2020; Zimmer et al., 2020). In addition, most of these systems allow to distinguish between wet and dry conditions, without any additional information on the flow magnitude. To improve identification of the causes of intermittency and to expand monitoring capabilities, it is imperative to adopt reliable equipment that may provide comprehensive observations of the spatial patterns of flow regimes. In this vein, visual inspections are the ultimate test to detect the flow status (dry vs wet) in streams. Unfortunately, in-person observations are often time and labor intensive, and may be challenging in difficult-to-access environments. On the other hand, remote sensing has been proposed as an efficient approach to monitor large areas with spatially continuous and frequent coverage (Borg Galea et al., 2019).

Cameras have the potential to afford remote and continuous observations of relatively extended areas without affecting the flow field. Therefore, optic systems are increasingly being installed in existing monitoring stations to complement information gathered with and to assess the performance of traditional equipment (Pagano et al., 2020; Vetra-Carvalho et al., 2020). Portable and permanent implementations of image-based systems have demonstrated the suitability of photogrammetric approaches to characterize the surface flow velocity field and monitor floods in natural streams (Tauro et al., 2014; Perks et al., 2016; Tauro et al., 2016a,b; Pearce et al., 2020). With regards to water level detection, in Kaplan et al. (2019), time-lapse photography has been adopted to show the absence or presence of flow in ephemeral and intermittent streams through image processing.





Besides the detection of the flow status, imagery can be effectively used to quantitatively estimate water level. For instance, in industrial applications, optics-based non-contact techniques are routinely constructed to estimate the depth of liquids in tanks (Chakravarthy et al., 2002; Reza and Riza, 2010; Yu, 2014). In river systems, imagery has opened novel capabilities toward water level measurement in ephemeral settings (Schoener, 2018). Since the pioneering work by Takagi et al. (1998), most of these image processing approaches rely on the detection of the water line, that is, the "boundary" sub-horizontal line

at the water-air interface. In Takagi et al. (1998), the appearance of the water line is highlighted through the inflection point on a metal measuring board. A similar approach is described in Kim et al. (2011), whereby the borderline between the water and a ruler is automatically detected through the histogram of consecutive images. In Sakaino (2016), a procedure based on the histogram is also applied to detect a flooding event. Staff gauges of uniform color are proposed in Royem et al. (2012) to enhance the visibility of the waterline in pictures captured with a low cost digital camera. Therein, a color conversion enables

image segmentation and identification of the staff against the water.

     Image-based approaches generally consist of the following steps: extracting grayscale images from digital cameras, applying image binarization through user-defined thresholds, conducting morphological operations to simplify images, and then detecting the water line through edge detectors (Shin et al., 2008; Yu and Hahn, 2010; Gilmore et al., 2013; Lin et al., 2013; Yang et al., 2014; Lin et al., 2018). Image distances in pixels are related to the real world reference system by using fidu-

cials. Another approach for shore line detection that does not require the presence of measuring boards or staffs in the field of view encompasses a spatio-temporal texture analysis on a sequence rather than a single image (Kröhnert, 2016; Kröhnert and Meichsner, 2017). This technique has proved particularly suited for implementation on smartphone technology (Elias et al., 2019). Further, combination of this method with high-resolution topography from structure from motion has enabled accurate water stage estimations in ungauged catchments (Eltner et al., 2018). Further computer vision methods for water level detection

include semantic segmentation algorithms (Lopez-Fuentes et al., 2017), analysis of the water intensity signal relative to fixed features (Leduc et al., 2018), principal component analysis (Young et al., 2015), and machine learning (Chapman et al., 2020).

     The pervasive use of digital cameras as well as the availability of high-performance digital systems at affordable costs make image-based measurements a promising approach to overcome the bottleneck of streamflow data scarcity in headwater streams. On the one hand, large volumes of image data, such as those available through social media posts, can be processed with deep

learning algorithms to extract information on the water depth (Chaudhary et al., 2020; Feng et al., 2020; Jafari et al., 2020; Lin et al., 2020). On the other hand, crowdsourcing and citizen science initiatives aimed at collecting and processing imagery may be viable alternative approaches to traditional methods to monitor the stream status (Seibert et al., 2019; Nardi et al., 2020). In fact, public involvement may lead to temporally dense data at a multitude of locations. Further, citizen-supervision may be highly instrumental to develop benchmark datasets in ungauged sites.

Despite their promise, autonomous photogrammetric methodologies have been sparsely adopted for hydrological measurements. For instance, a proof-of-concept discharge measurement is demonstrated in Stumpf et al. (2016), whereby the water level is derived by projecting the digital terrain model on the image plane. In Nones et al. (2018), detection of the water line displacement provides insights on long-term fluvial morphodynamic processes. In Ridolfi and Manciola (2018), unmanned aerial vehicle technology and water line detection are combined to compute the water level at a dam site. Near infrared (NIR)-





imaging from a video surveillance system is proposed in Zhang et al. (2019a,b) to measure the water level at large scale riverine
sites in complex illumination settings.

Motivated by latest advancements of digital cameras and computer vision techniques, in this work, we design and develop a
stage-camera system to monitor the water level in intermittent and ephemeral streams in ungauged headwater areas. Different
from approaches that rely on measurement boards, this system seeks to estimate the water depth from the out-of-water length

of a simple white-painted steel pole. Images of the pole are captured with a low cost wildlife camera set in the time-lapse
mode that enables image acquisition both during the day and at night. Images are off-line processed through a computationally
inexpensive algorithm featuring image quantization based on automatically determined intensity thresholds and image bina-
rization. The actual length of the pole is a priori known and, thus, camera calibration through the acquisition of fiducials can
be circumvented. The feasibility of the stage-camera for hydrological measurements is demonstrated through a set of experi-

ments performed in controlled settings and in an actual small scale catchment. The objectives of this work are: i) to describe
an efficient yet affordable system for water level monitoring in small-scale streams; ii) to develop a simple procedure to accu-
rately estimate water levels from images in a wide range of illumination and meteorological conditions; and iii) to demonstrate
continuous non-intrusive water level monitoring under a broad range of field conditions. Despite these efforts, the potential of
stage-cameras for capturing flow patterns in headwater streams still needs to be better explored.

## 2   Monitoring framework

### 2.1   Stage-camera system

The system comprises a consumer grade wildlife camera, the Trap Bushwacker D3, and a white-painted steel pole, see Figure 1.
The camera captures $16\,\mathrm{Mpix}$ images in the time-lapse mode from $3\,\mathrm{s}$ to $24\,\mathrm{hr}$ time intervals. At night, high image visibility is
afforded through shooting in the near infrared (NIR) band ($850-940\,\mathrm{nm}$). Power loading is enabled through 8 rechargeable AA

batteries. The pole is $8\,\mathrm{mm}$ in diameter and $1.5\,\mathrm{m}$ in length and its uppermost end is clearly visible through a black stripe. Once
installed in the stream thalweg, the pole's small thickness yields minimal resistance to the flow. The system costs approximately
€185 (the Trap Bushwacker D3 is available on the market at €110).

In this implementation, the camera is installed on a river bank at a few decimeters from the riverbed, see the left panel in
Figure 1, with its axis roughly perpendicular to the longitudinal section of the river bed. Images are taken every 30 minutes,

thus guaranteeing a system runtime of approximately two weeks, and then stored in a $32\,\mathrm{Gb}$ SD card.

### 2.2   Image processing

Image processing consists of two phases. First, the out-of-water pole length is estimated through an image-based algorithm.
Then, raw measurements are filtered through a simple statistics-based scheme. These two phases are detailed in what follows.





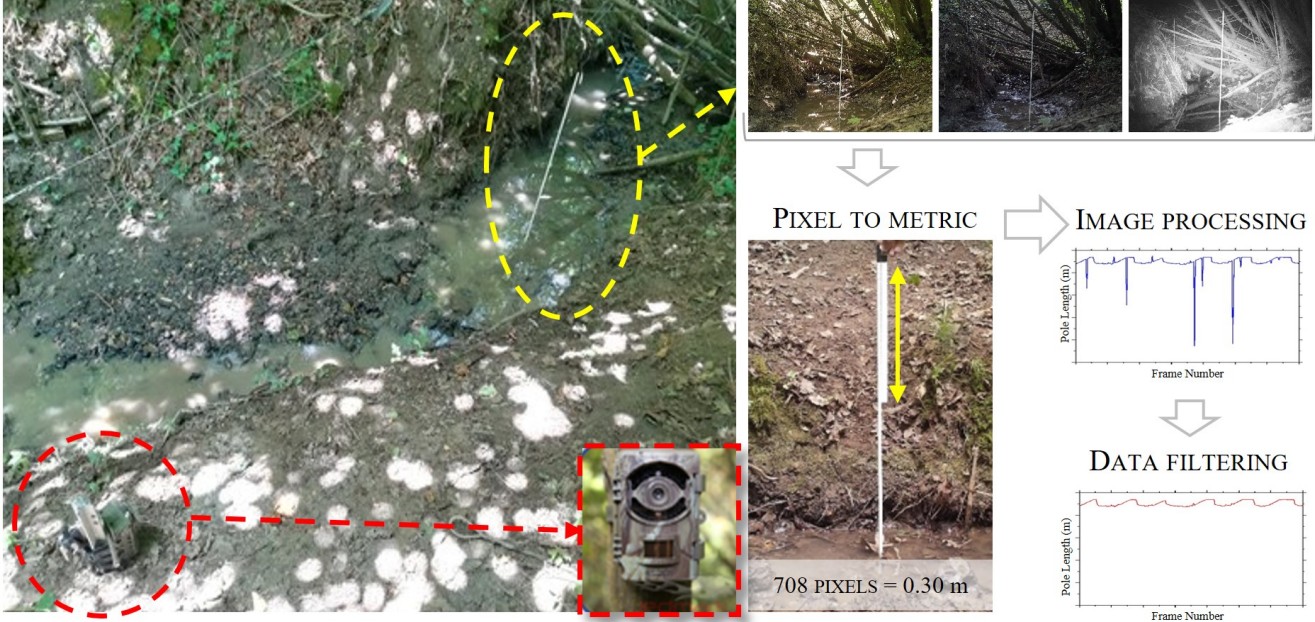

**Figure 1.** Left, stage-camera system including a wildlife camera (enclosed in the red dashed circle) and a white-painted steel pole (enclosed in the yellow dashed ellipse). In experimental tests, the camera is installed on a stream bank with its axis roughly perpendicular to the longitudinal section of the river bed. Right, images are captured both during the day and at night. Grayscale images are segmented and binarized to detect the pole. Then, a filtering procedure removes eventual outliers. Pixel to metric conversion is conducted by preliminarily calibrating images in situ based on the pole length.

### 2.2.1 Water level estimation

The water level estimation procedure relies on the assumption that the pole is the brightest object in the field of view. Colored images are converted to grayscale by eliminating the hue and saturation information while retaining the luminance. Then, a region of interest (ROI) larger than the pole is automatically trimmed around it, Figure 2. Images are segmented through the nonparametric unsupervised Otsu's method based on gray-level image histogram (Otsu, 1979), and then quantized according to the classes assigned in the segmentation. The number of segmented classes is set according to illumination conditions:

in case of diurnal images, images are segmented in two (three or five) classes if they display homogeneous (heterogeneous) backgrounds; conversely, images captured at night are segmented into seven classes. Image binarization is then performed by setting the brightest class to white and the darker classes to black. To prevent eventual ambiguities due to sunlight reflections, the class pertaining to the pole is identified by imposing a constraint on the admissible number of pixels (that is, the amount of pixels pertaining to the pole spans within an a priori defined range). Among 8-connected elements depicted in binary images,

the pole is identified as the object with the largest number of pixels. Hence, it is bounded in a rectangle and, depending on background complexity, either the side or the vertices of the bounding box are used to estimate the out-of-water pole length.





Water level is then estimated by subtracting the out-of-water length from the total pole length. Pixel to metric conversion is conducted by preliminarily calibrating images in situ based on the pole length.

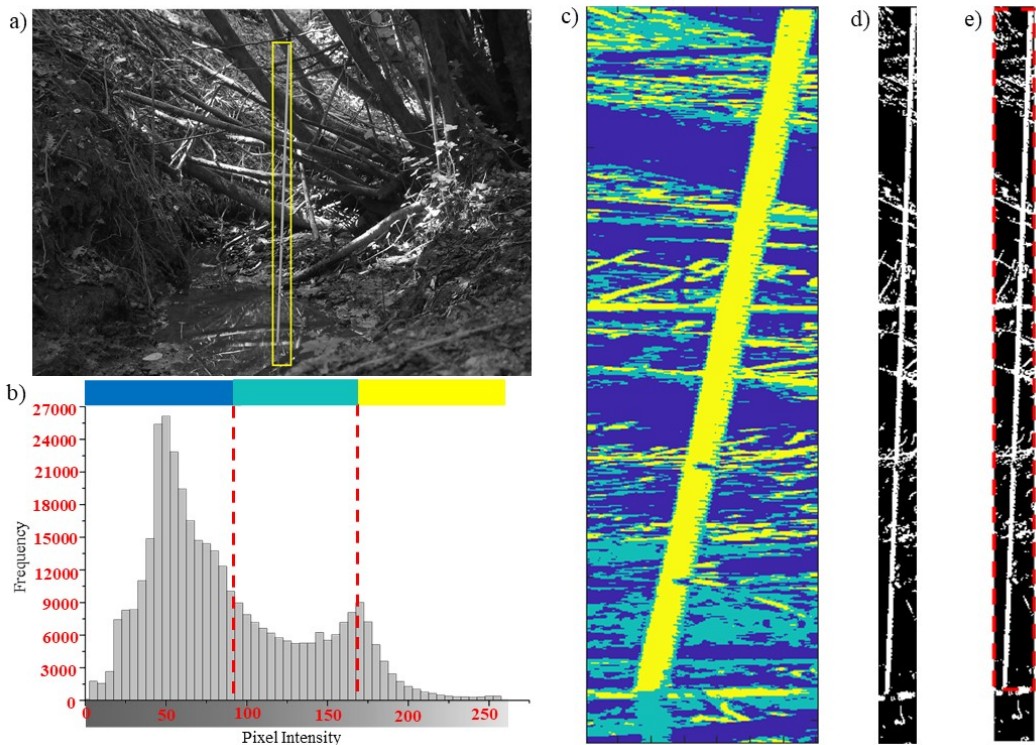

**Figure 2.** Image processing scheme: a) a region of interest is established in grayscale pictures, b) segmentation through Otsu's method (Otsu, 1979) is executed, c) images are quantized in classes (three classes are herein shown), d) image binarization is performed, and e) a red dashed bounding box encloses the pole.

### 2.2.2 Data filtering

Raw water level estimations are further processed through a filtering procedure to remove eventual outliers, Figure 3. Outlying data may eventually be found due to adverse illumination conditions that yield inaccuracies in detecting the pole in images. First, a moving average is computed and subtracted from raw data, $\epsilon$ in Figure 3c). Outliers are defined as data whose absolute difference exceeds a threshold set to the 90% quantile. Such data are removed and replaced with inputs obtained through linear interpolation between values acquired at the previous and subsequent time steps. When executing the moving average, the

window width is set to three values to maximize the difference between raw and averaged data only on those records that are strongly over- or underestimated. At the same time, this minimizes the difference between raw and average data close to errors, and thus, values immediately before and after outliers fall within the 90% quantile and are not removed.



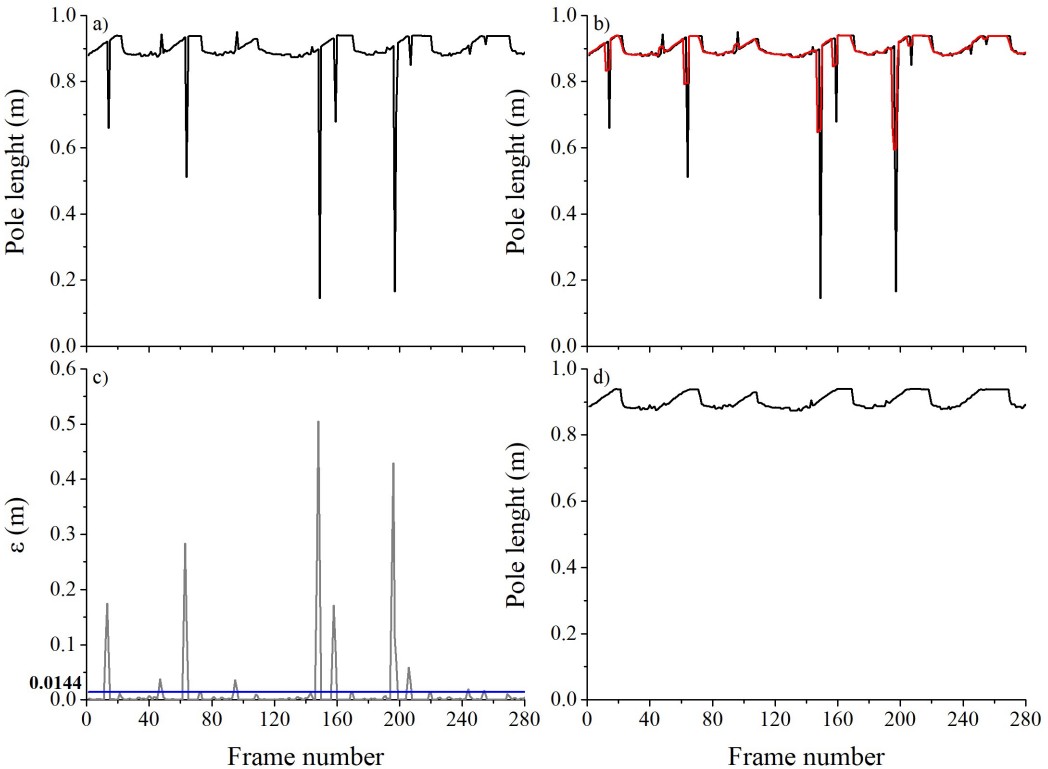

**Figure 3.** Image filtering scheme: a) time series of raw out-of-water pole length data, b) data processed through a moving average (red), c) outliers are identified as data above the $90\%$ quantile (horizontal blue line), and d) output filtered water levels.

## 2.3 Experimental tests

Five experimental tests with an increasing degree of complexity are carried out to evaluate the efficiency of the stage-camera,
see Figure 4. Two preliminary tests are performed in controlled conditions to assess the efficacy of the water level detection
algorithm to estimate the pole length. Specifically, a set of tests (test A) is executed in controlled light and image background
conditions in an external courtyard at the University of Tuscia, Viterbo, Italy. More challenging illumination and backgrounds
are tested in the outdoor MecHydroLab terrain parcel at University of Tuscia (test B). Finally, three additional tests (tests C to
E) are executed in a real stream in the Montecalvello catchment ($20\,\mathrm{km}$ far from Viterbo). At the cross-sections of tests C to E,
the catchment drainage area is equal to $1.9\,\mathrm{km}^2$. The stream bed is approximately $1\,\mathrm{m}$ wide, and the water level is up to $10\,\mathrm{cm}$
deep.

Test A encompasses homogeneous light and background conditions, Figure 4a. Namely, the pole is clearly illuminated by
diffused sunlight in the foreground of a black panel. The pole is set in a tank where quiescent water is maintained. In such
settings, the effects of water reflections are minimized. A total of 65 images collected from March 16th to 20th, 2020, are
analyzed.



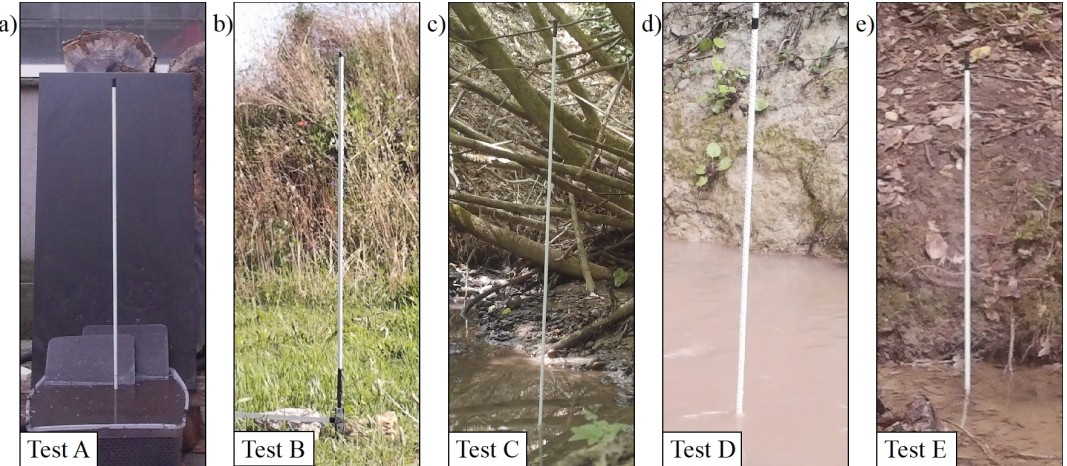

**Figure 4.** Experimental test conditions: a) controlled light and image background conditions (test A), b) natural illumination and background (test B), c) variable light conditions, heterogeneous backgrounds, and stream level fluctuations (test C), d) intense direct light during most of the day (test D), and e) heterogeneous backgrounds, irregular illumination, and raindrops (test E).

In test B, both the camera and the pole are set on a tripod and heterogeneous background and variable light conditions are introduced, Figure 4b. Specifically, the pole is hit by both diffused (at sun dawn and sunset) and intense light (during the rest of the day). Diverse illumination along the pole yields complex images. A total of 90 images collected from June 23rd to 25th, 2020, are analyzed.

Test C is executed in the Montecalvello catchment, Figure 4c. The camera is installed at a stream bank and captures the pole set in the thalweg. Collected images encompass variable light conditions, heterogeneous backgrounds, and water level fluctuations. The site features diverse illumination conditions: diffused light hits the pole early in the morning and late in the afternoon. During the day, sunflecks (see, for instance, the bright spots on the left of Figure 1) occur and are scattered throughout the field of view due to irregular riparian vegetation canopy. Daily fluctuations in the water level lead to a dry riverbed in the afternoon. A total of 286 images collected from June 2nd to 8th, 2020, are analyzed.

Test D is carried out approximately $2\,\mathrm{km}$ downstream of site C, Figure 4d. Therein, riparian vegetation canopy is absent and intense light hits the field of view during the day, whereby diffused light occurs early in the morning and late in the afternoon. A total of 28 images collected from October 14th to 15th, 2020, are analyzed. Towards the end of the test (on October 14th, from 6 pm to 10 pm), a storm event occurs (average rainfall intensity equal to $5.4\,\mathrm{mm/h}$ and 4 hours long), which is captured in analyzed data.

Finally, test E is executed in a river segment close to the site of test C in more adverse hydrometeorological conditions, Figure 4e. Specifically, 201 images are collected from March 2nd to 6th, 2020, during rainfall events. This case represents critical experimental conditions, where heterogeneous backgrounds combined with irregular illumination and raindrops sensibly affect image quality. In particular, moisture on the camera and mud on the NIR sensor tend to yield noisy images.



## 2.4 Data validation

Water level data obtained from the automated image processing methodology are benchmarked against values gathered through manual inspection of the pictures. For each experimental test, image sequences are visually analyzed and the pole-water interface ($y_{\text{int}}$) identified by eye. The uppermost end of the pole ($y_{\text{top}}$) is determined once for each image sequence, and the actual pole length is estimated from $|y_{\text{top}} - y_{\text{int}}|$. Water levels are finally computed by subtracting such actual length from the pole's a priori known total length. The length in pixel is computed from the pixel to metric conversion coefficient introduced in Section 2.2.1.

## 3 Results and discussion

The mean absolute error (MAE) between image-based water levels and supervisedly estimated benchmark data as in Section 2.4 are between $0.29\,\text{cm}$ ($0.26\,\text{cm}$) and $2.20\,\text{cm}$ ($2.02\,\text{cm}$) for the unfiltered (filtered) water levels, Table 1.

**Table 1.** Mean absolute error (MAE) values between image-based water levels and supervisedly estimated benchmark data for each experimental test. Values are computed both on unfiltered (unfilt.) and filtered (filt.) water levels.

| Test | MAE (unnfilt.) | MAE (filt.) |
|------|------|------|
|      | cm   | cm   |
| A    | 0.29 | 0.26 |
| B    | 1.07 | 0.57 |
| C    | 1.34 | 0.33 |
| D    | 1.39 | 1.54 |
| E    | 2.20 | 2.02 |

As expected, test A exhibits the closest agreement between image-based and benchmark data (lowest MAE values in Table 1). As experimental complexity increases, larger discrepancies between estimated and reference water level values are found. In test B, a full agreement is observed with the exception of a few entries, Figure 5. In this case, application of the filter corrects most of such records and decreases the MAE by about 47%. Interestingly, in test C, the reconstruction of water level provided by image processing clearly describes water fluctuations, Figure 5. Unfiltered data show a few strongly underestimated water levels (leading to large MAE values) in the rising limbs of the time series. Such outliers are effectively removed and corrected with the filtering procedure. Remarkably, this abates the MAE by about about 75 %.

The storm event occurred on October 14th is accurately captured in both unfiltered and filtered water levels in test D, Figure 5. However, severe illumination from 2.10 pm to 3.30 pm leads to level overestimations during this period. Unfortunately, application of the filtering procedure did not remove such inaccuracies, thus leading to a higher MAE value than for unfiltered data. Finally, test E shows the highest MAE values due to challenging rainfall conditions, Figure 5. Poor image quality leads to



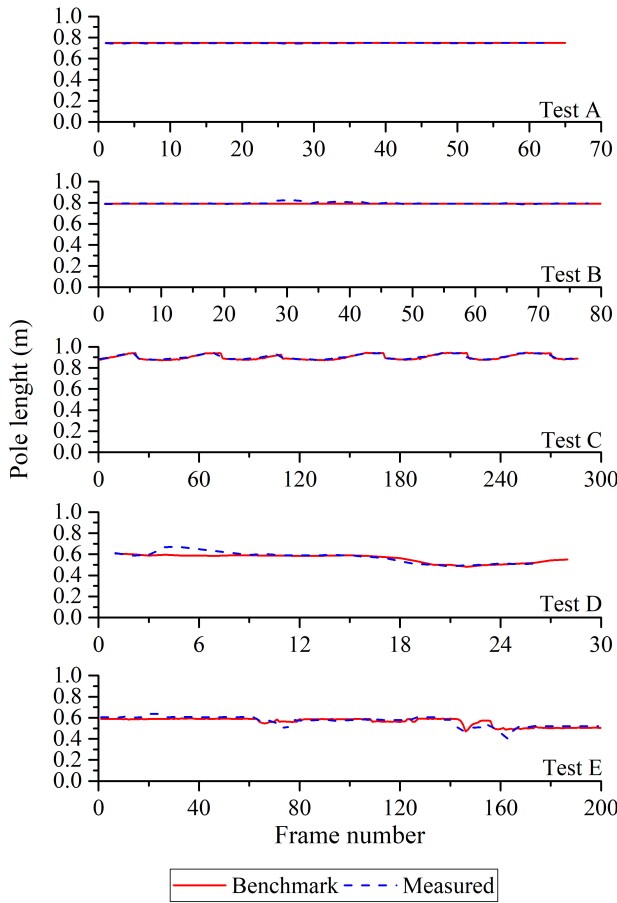

**Figure 5.** Experimental results: from top to bottom, the time series of filtered water levels (blue) are compared against benchmark values for all experimental tests (red).

some underestimations in the water level, with maximum discrepancies on the order of more than 10 cm. Notably, application of the filtering procedure reduced the MAE by 8%.

Water level estimations with the stage-camera system are generally in agreement with benchmark data and the use of such methodology is promising for monitoring intermittent and ephemeral streams in cases where other technologies for discharge monitoring are not viable options. Future ameliorations may help in further enhancing the performance of the approach. First, illumination conditions and the presence of sunflecks are important controls on image quality and severely affect the estimation of the out-of-water pole length. Currently, no image is discarded before image processing irrespectively of eventual image intensity saturation regions. In future implementations, preliminary controls on the global brightness of single images may be conducted to inform the application of specific enhancement procedures or a more advanced selection of the number of segmentation classes. Alternatively, the setup may be integrated with luminance sensors to guide image processing.





Our analysis indicates that rainfall has a high impact on image quality. The presence of raindrops leads to images with unsharped edges, making challenging the identification of the pole. This issue may be mitigated by covering the camera and eventually tilting it to prevent raindrops to spill onto the camera objective. In this latter case, introducing an orthorectification phase before image processing may be required. Regardless of the specific hydrometeorological conditions, pole visibility can

be further enhanced by replacing the pole with a wider board and/or a board of an alternative uniform color. These modifications may help in unambiguously emphasizing the board as the brightest object in the field of view. On the other hand, increasing pole dimensions may also lead to large quantities of suspended material stacking around it on the stream surface. This will affect the image intensity of the region in proximity of the pole-water interface. In our experiments, stacking of suspended material is observed only for test E; however, this leads to minimal overestimations of water levels during the recession of the

rainfall event, which are partially compensated by the filtering procedure. In future experiments, installing a pole (or board) upstream of the measurement site is expected to help in intercepting the suspended material before its transit across the camera field of view.

In some cases (e.g. test D), the filtering procedure is not fully successful in removing water level overestimations. This is due to the fact that the moving average window width is minimized to efficiently identify the outliers. When image-based

water levels are not greatly overestimated though, it may be difficult to detect eventual outliers. For instance, in test D, the presence of a sequence of slightly overestimated records causes higher discrepancies beetwen averaged and accurate raw data than between averaged and overestimated records. Thus accurate records are replaced with higher values, and the MAE on filtered data increases. Increasing the camera acquisition frequency, thus better delimiting erroneous records, may be a strategy to compensate for such an issue. In turn, increasing the frame frequency has implications for on site image storage and the

overall system energy consumption. To this end, future efforts will be devoted to optimize the stage-camera setup and software components towards the development of an autonomous sensing platform. Specifically, solar panels may be installed to enable longer runtimes and on site image processing through an embedded computing unit. Low power embedded systems present several advantages for environmental monitoring in remote environments (Tosi et al., 2020). Also, they open novel avenues on the scalability of such remote measurement approaches. By interfacing an agile stage-camera platform with a wireless

infrastructure, it may be possible to directly transmit water level data in real time to a master system, thus circumventing the need for on site inspections and greatly simplifying data acquisition and processing.

## 4    Conclusions

In this paper, a cost-effective remote approach was developed to estimate the water level of ephemeral and intermittent streams. The system comprises a wildlife camera and a reference pole of known length. A simple image-based processing algorithm was

developed and combined with a filtering procedure to compute water level data in complex natural settings. In the presented implementations, the stage-camera captured images every 30 minutes and featured an autonomy (of both battery and data storage) of two weeks. Stage-camera data were in general agreement with benchmark values (maximum mean absolute errors around 2 cm in severe hydrometeorological conditions) and the filtering procedure was effective at identifying outlying data.

Mean errors observed in different tests ranged within $0.26 - 2.20\,\mathrm{cm}$, suggesting a good potential of this approach for stage
monitoring.

Compared to traditional monitoring methods, the stage-camera offers several advantages: it allows for simultaneously estimating the water level in headwater streams and for supervising the stream area and banks. Data are derived without deploying any sensors in the flow and minimal maintenance is required through in-person inspection of the equipment. Also, in upland areas, small fluctuations in the water level correspond to significant flow discharge variations, and, therefore, the accurate and continuous observations enabled by stage-cameras may be highly beneficial to map flow intermittency. Finally, the system is inherently designed for enabling accurate and continuous observations at multiple locations in ungauged areas of natural catchments. Therefore, we believe that stage-cameras may be valuable additions to the toolkit available to experimental hydrologists and environmental practitioners.

*Code and data availability.* The datasets and codes generated for this study are available on request to the corresponding author.

*Author contributions.* SG was responsible for conceptualization. FT developed the image processing code, and SG and SN developed the data filtering procedure. SN performed the experiments and formal data analysis. SN, CA, AP and SG developed figures. FT prepared the manuscript with contributions from all co-authors. GB and SG were responsible for project administration and funding acquisition.

*Competing interests.* No competing interests are present.

*Acknowledgements.* This study was supported by the European Research Council (ERC) DyNET project funded through the European Community's Horizon 2020 - Excellent Science - Programme (grant agreement H2020-EU.1.1.-770999) and by the Italian Ministry of the Environment, Land and Sea (MATTM) through the project "GEST-RIVER Gestione ecosostenibile dei territori a rischio inondazione e valorizzazione economica delle risorse". Flavia Tauro acknowledges support from the "Departments of Excellence-2018" Program (Dipartimenti di Eccellenza) of the Italian Ministry of Education, University and Research, DIBAF-Department of University of Tuscia, Project "Landscape 4.0 – food, wellbeing and environment".



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
