# Peer review of "Technical Note: Low cost stage-camera system for continuous water level monitoring in ephemeral streams"

_Hydrology and Earth System Sciences, 2021_

## Author Comment (AC1)

**AUTHORS' REPLY TO:**

- Community Comment, CC, by Kenneth Chapman;
- Referee Comment 1, RC1, by Nils Kaplan;
- Referee Comment 2, RC2, by Anonymous Referee;

In the following we report in "black" the CC, RC1, RC2 comments, in "blu" the authors' replies and in "red" the authors' actions that will be implemented in the revised version of the manuscript.

**COMMUNITY COMMENT - Kenneth Chapman**

Just want to post a note on your really nice work. We have been working on this problem for about a decade now and your innovation of using a pole and an "in the water--out of the water" approach to measuring water in the settings described in your paper is very nice and novel in our experience. This is significantly different from using a calibration target (our method) or with a staff gauge. Its power is the ability to install something really simple in a stream and still get measurements that are accurate enough to be of interest with fairly simple image processing algorithms and calibration techniques. The balance between simplicity and ease of use vs. precision of measurement is a difficult one. I am looking forward to following your work.

*Authors' reply.* We are particularly glad and flattered by Kenneth's comment that further motivates us on this research field.

**REFEREE COMMENT 1 - *Nils Kaplan***

General comments:

This paper presents a promising re-interpretation of an image-based water level monitoring system for intermittent rivers and ephemeral streams (IRES) using a consumer wildlife camera with integrated time-lapse function. The continuous image-based monitoring of water level adds valuable information of the temporal dynamics of flow in IRES that were traditionally only monitored sporadically during field campaigns or with intermittency or EC sensors. However, those sensors cannot provide the visual information of an image, which allows for the evaluation of automated water level detection, the presence/absence of water in the channel or clogging of the stream channel. The presented method has a high potential to support future monitoring campaigns in the IRES research. However, some minor corrections and supplementary information needs to be added to the manuscript. Thus, I recommend the publication after minor revisions of this manuscript.

*Authors' reply.* We are glad that Nils Kaplan appreciates the proposed method considering it as an approach with high potential to support future monitoring campaigns in the IRES research.

Specific comments:

Line 10: What are the "severe hydrometeorological conditions"? Only strong precipitation events or also extreme dry conditions? Please clarify.

*Authors' reply*. We agree with the reviewer that we were too general. Specifically, we refer to conditions present during thunderstorm with presence of rainsplash on the camera support and with high density of drops and fog in the frames.
*Authors' action.* We modified the text better specifying the severe conditions.

Line 23: […]"uppermost" catchment "areas"

*Authors' action.* Thanks, we accordingly modified the text.

Line 23: "posit" seems to be an unusual wording in this context

*Authors' action.* Thanks, we accordingly modified the text.

Line 26: I suggest to use "Conventional" in stead of "Traditional"

*Authors' action.* Thanks, we accordingly modified the text.

Line 35: The citation of Kaplan et al., 2019 is a bit misleading as the presented dataset achieves it spatial resolution of an combination of EC-sensors, time-lapse imagery and conventional gauging.

*Authors' action.* Thanks, we better specify it in the text.

Line 38: I would not recognize the noise of the sensor as biggest thread for EC-data accuracy for presence/absence of water but the accurate position of the sensor at the deepest point in the channel cross-section. The noise introduced by clogged material at the sensor is rather well to handle by using a little large thresholds for the EC-values.

*Authors' action.* Thanks, we agree, we accordingly modified the text in line 38.

Line 45: is very close to the sentence in Line 1. Maybe consider small changes to the wording.

*Authors' action.* Thanks, we accordingly modified the text.

Line 110: I experimented with a similar setup using a white pole, but the reflection of the pole through a clear water surface were too bright to extract the water line. Thus, I am interested in the paint you used for the pole (special matt color used?) and if the paint had any ability to prevent growth of algae, which would potentially affect the image processing algorithm. This would add valuable information to the manuscript.

*Authors' reply.* We agree that the pole color could be pivotal in the proposed method. Presently, we are investigating on the best possible option exploring the RGB channel for different colors. In the described experiments we adopted the "OBI Spray Colour Pure White RAL 9010, Opaque" and we did not see growth of algae.
*Authors' action.* Thanks, we added the paint information into the text.

Line 112: A little more details on the mounting system would be beneficial. Were specific measures be taken to prevent theft and/or camera movement?

*Authors' reply.* We sadly agree with Nils Kaplan, indeed we did not plan a specific measure to prevent theft, which happened just few days ago.

Line 119: In this section a note on programming language and potentially packages used to write the image processing software would be great.

*Authors' action.* Thanks, we added the language and software information into the text.

Line 122: Is the ROI static or set by an algorithm? Is the algorithm capable to respond to issues with movement of either the pole or the camera?

*Authors' reply.* The ROI is static and presently the algorithm is not capable to respond to such issues. This is subject of ongoing work.

Line 124: Is the number thresholds individually calibrated for each site or automatically set according to the illumination conditions?

*Authors' reply.* Presently, we are investigating the optimal number of thresholds and its stability. For the paper application we did not calibrate or optimize the number of thresholds, we only selected a preliminary reasonable value.

Line 137: Please add the information about the size of the moving average window already in this sentence to avoid confusion.

*Authors' action.* Thanks, we accordingly modified the text.

Line 138: […] "difference" between moving average and raw value […]?

Line 138: […] "set to the 90% quantile" of the moving window.

Line 138: The 90% quantile threshold might be a little bit too low to capture the fast dynamics of extreme events in ephemeral streams.

*Authors' Reply to the last three comments (Line 138).* We agree that the text was not clear. The 90% quantile is referred to the difference between moving average results and raw values. Values higher than this threshold are removed and filtered.
*Authors' action.* We clarified the text.

Line 163: May consider to use "Light scatter" or "Scattered sunlight" instead of "sunflecks".

*Authors' action.* Thanks, we accordingly modified the text.

Line 182: Due to the specific dedication of the method to monitoring IRES it would be very beneficial to add an analysis for the MAE for the dry states of the channel compared to the flowing conditions. In case only the site "C" had dry conditions, it would help already to add this.

*Authors' reply.* Yes, thanks for the useful suggestion. We are presently developing an accurate analysis on the different flow regimes that will be submitted in a future paper.

Line 195: Many image sequences during severe rainfall events were acquired as NIR images in the study of Kaplan et al., 2019. Thus, the difference of MAE between RGB and NIR images might be also an interesting information to add (they might be a reason for the higher MAE here).

*Authors' reply.* We appreciate this comment, indeed we are investigating on that and it will be included in a future paper. We confirm that during the day and at night performances are different. The MAE is very low with NIR images and relatively high with RGB. Presently, we are comparing different supports and colors.

Line 202: Simple image saturation statistics might already be sufficient to remove some of the blurriest images before the actual analysis.

*Authors' reply.* We agree, indeed we are investigating also on preprocessing frames in order to optimize the postprocessing filtering.

Line 210: From an image processing point of view a white pole should be the brightest object compared to other colors. However, the difference between glossy vs. matt paint could be interesting.

*Authors' reply*. We appreciate the suggestion. Concerning colors we can anticipate that "red" could provide lower MAE.

Line 211: Additionally, to the debris that could get stuck at larger poles, they may have also a larger potential to get eroded.

*Authors' reply.* We agree, potentially it could, however, fortunately, we did not observe relevant erosion phenomena.

Line 241: The advantages of the system could also be stated earlier; potentially at the end of the introduction

*Authors' action.* Thanks, we accordingly modified the text.

Line 242: "with minimal flow disturbance through the pole" instead of "without deploying any sensors in the flow"
*Authors' action.* Thanks, we accordingly modified the text.

Technical corrections:

Line 223: "frequency of time-lapse image acquisition" or "image acquisition frequency" instead of "camera acquisition frequency".

*Authors' action.* Thanks, we accordingly modified the text.

Line 224: "time-lapse interval" instead of "frame frequency"

*Authors' action.* Thanks, we accordingly modified the text.

Figures:
I suggest to include a figure describing the processing algorithm in a flow chart.

*Authors' action.* Thanks, we have included a flowchart in the manuscript.

**REFEREE COMMENT 2  -** Anonymous Referee
The application of a camera approach as a low-cost system for water level observations is certainly interesting. However, my rather fundamental issue with this manuscript is the added value of this approach. These days, one can buy water level sensors for 100-200 USD which provide measurements with a millimetre resolution. As the described system utilizes a pole installed in the stream (which is a neat idea), there is also not the argument that no in-stream installations are needed for the camera approach. So, as cool as the described approach is, I am not sure about its practical value. At the end of the manuscript, the authors mention the importance of having pictures of the streambed. I could agree, but if this is the added value, it should be addressed before and in more detail.

*Authors' reply.* We are glad that the reviewer considers the proposed approach interesting, at the same time we respectfully disagree on the absence of stage-cam added value. The presently available sensors for water level observations could have significant drawbacks in a variety of practical applications that could be overcame by the not-intrusive proposed monitoring system. In ephemeral streams, for which the availability of continuous water levels is crucial, we usually have challenging conditions. Indeed, water is absent for a long time and the flood hydrographs could be very fast with rapid and particularly turbulent flow rich of sediment transport, making impossible to install intrusive systems. Moreover, even minimal sediment transport can cover sensors eventually

installed on the river bed, thus inhibiting permanent installation. Finally, ephemeral streams are typically located in inaccessible areas rich with vegetation, without electrical power, and without enough solar irradiance for installing solar panels. So, other classes of common water sensors cannot be easily installed (or they are economically unfeasible) due to their power requirements. In fact, in our experimental site we tried pressure transducers without success.

My other major concern is the study design. First of all, I am afraid I have to disagree that manual inspection of the images should be the sole comparison. Here a fully independent approach should be used, i.e. a 'real' water level sensor. Second, the observed level variations (Fig 5) are really small. For evaluation, there should be larger changes, especially also in the 'lab setting' of Test A. Using a constant level here limits the evaluation.

*Authors' reply.* We understand the point of the reviewer and we tried alternative sensors, however we realized that the best benchmark is the manual image analysis for quantifying pole length. Actually, we realized that such approach would be a good benchmark also for validating common water level sensors, indeed it is only minimally affected by errors, which are lower than those typically introduced by every sensors. Concerning level variation, we illustrated different kinds of events and, being the present manuscript a technical note, where the method is proposed for the first time, we thought that providing results for static water levels where basic and specific conditions are tested would be appropriate.

Approach
(fig 2): wouldn't it be advantageous to rotate the image so that the pole is exactly vertical? I am not sure I understood how the tilting of the pole, in reality, is considered. Please clarify.

*Authors' reply.* We agree with the reviewer that we were not clear enough in describing the algorithm. Actually, the measure consists in identifying the vertical height of the bounding box so the pole inclination determines a negligible error (i.e. the error due to the inclination for the frame in Figure 2a is equal to 0.7 mm).

*Authors' action.* we have included a flowchart of the algorithm in the manuscript.

Fig 4: why is the pole so long? This seems to make things rather unstable

*Authors' reply.* The pole length is not an issue, it depends on the expected maximum water level that the stream could reach during an extreme event. We did not observe instability since the pole is inserted in the ground for 40-50 cm.

*Authors' action.* we better specified in the text the pole installation description.

Minor comments:
Sometimes long lists of references are given, e.g. P1L18-19, please try to be more specific about the contributions of the individual papers.

*Authors' reply.* We thank the reviewer, this long list is to underline how crucial the water level observation is for hydrological investigations. Providing details would make the introduction too long including off topic information.

P2L41: here 'only' should be added for clarification

*Authors' action.* Thanks, we accordingly modified the text.

Often hydrologists are interested in flows rather than in levels. Please comment on the use of level data without a rating curve.

*Authors' reply.* We agree that usually flows are the final aim of hydrometric observations, however, as mentioned in the Introduction, the water level is, per se, crucial information for a variety of studies. A system providing flow observation in such challenging conditions is the subject of ongoing work.

I would prefer to have results and discussion in two separate sections for better readability.

*Authors' reply.* We agree that usually it is appropriate to keep separate results and discussion, however in the present manuscript these two sections would be too short with possible repetitions and redundant information.

---

## Author Comment (AC2)

**REFEREE COMMENT 2 - Anonymous Referee**

This manuscript presents a method to automatically measure water level in streams using NIR-cameras and image processing techniques. The paper is generally well written and the results are promising. However, I do think that the paper should be improved before it can be considered for publication in HESS. First, the structure of the paper is somewhat unbalanced. The introduction is relatively extensive, making the results and discussion section seem rather marginal. I think the latter section would benefit from a stronger and more elaborated evaluation of the presented method, and include a more detailed outlook to future work and potential applications. Second, the data availability statement is not in line with HESS policy. This should be updated before the manuscript can be considered. Finally, the paper would benefit from additional support for and clarification about the setup and choices made, detailed in the general and specific comments below.

*Authors' reply.* We are glad that the reviewer considers the proposed approach promising and we are grateful for the interesting suggestions for improving the manuscript.

**1. General comments**

**Introduction**

- Another reason why ephemeral streams are so relevant is perhaps that the onset of flow may result in the mobilization of (anthropogenic) debris and sediment as well?

*Authors' reply*. We agree with the reviewer
*Authors' action.* Thanks, we accordingly modified the text.

- The link with citizen science makes sense, as this offers an unprecedented opportunity for upscaling of data collection. However, how would this work for the locations of interested in this manuscript, i.e. ungauged headwater catchments? These may not be the locations where many citizens may be available to contribute with data collection.

*Authors' reply*. We agree with the reviewer, indeed we were not enough clear in specifying that we refer to citizen contribution for analyzing images (the manual inspection) in order to have a large benchmark database.
*Authors' action.* Thanks, we better clarified the text.

- The introduction in general is well-written. I do think it is a bit long and goes on a tangent here and there. Perhaps the authors can reduce the length a bit and focus more on the potential of their approach, and why this is a promising addition to the existing suite of monitoring techniques.

*Authors' action.* Thanks, we tried to reduce the introduction accordingly to the reviewer's specific suggestions.

**Methods**

- Perhaps a sketch of the monitoring setup can be included in addition to Fig. 1.

*Authors' reply.* we included it in the left side of Figure 1, where the camera and the pole are highlighted through the red and yellow circles.

What is the motivation for taking images every 30 minutes? What is the relevant timescale for ephemeral streams? I'd argue that a single to a couple of images per day would suffice, drastically reducing the required storage. With the current setup someone needs to read out the data every two weeks, which I would personally find quite much for ungauged headwater catchments.

*Authors' reply.* Unfortunately, a couple of images per day are, in general, not enough for an accurate reconstruction of the relevant stream dynamics. While the appropriate timescale might be case-specific, in our test catchment the ideal time resolution would be probably 5-10 minutes. In most headwater streams, the concentration time is very short and surface runoff can last for less than a few hours. In any case, the proposed methodology is quite flexible in terms of the selection of the appropriate temporal resolution, the only constrain being the post-processing time required for the image analysis. In lines 225-230 we describe the future work aimed at mitigating this potential drawback.

- 1: I find this figure a bit unclear. Perhaps some additional headings to complete the workflow makes it a bit clearer.

*Authors' action.* Thanks, we better clarified the caption of the Figure.

- Please include some more details about the setup. How long is the pole? How is the pole robustly placed in what looks to be a rather "wild" environment? What is the distance between the pole and the camera? How is the camera fixed? What is the estimated pixel length (mm, cm)?

*Authors' action.* Thanks, we better clarified the text.

- Maybe a overview map can be included to show the outdoor testing locations.

*Authors' reply.* As proof of concept that focuses on the proposed methodology, the location of the stage-cam could be not relevant, so we prefer to avoid additional figures that make the manuscript longer.

- For the data validation, was the water level identification done by the same person? Or by a group of people? If the latter, was there any bias between the observers? Also, I was wondering if there was a reason to not measure the water level with an accurate water level logger.

*Authors' reply.* For the case studies here described the validation was done by the same person. For other case studies that we are currently investigating we involved three people and computed the mean of obtained values. We did not observe a meaningful bias. As mentioned in the response to Reviewer 2, common water level loggers either fail or yield relevant errors due to the low water level and to sediment transport.

**Results and discussion**

- Why was Test A done with the same water level for each image? As this method is most valuable to detect changes in water level, would it not have been valuable to test the method for the full ranges of values?

*Authors' reply*. We included Test A as preliminary test for evaluating the proposed approach in the optimal conditions (homogeneous background and illumination).

- The method seems to work quite well for Test C, which includes quite some dynamic behaviour. For Test D and E, the dynamics seem not to be captured completely. Can the authors elaborate on this, including the implications for what that would mean for long-term monitoring?

*Authors' reply*. Thank you, in Section 3 we provide some comments on the performance of Test D and E.
*Authors' action.* we better clarified the text.

- The discussion is rather limited. I would encourage the authors to include a critical synthesis and more elaborated outlook on future work. What are the next steps for this method? How do the authors envision application in the field? Only for measurements of a couple of days, or also for seasonal or even multi-year monitoring efforts?

*Authors' action.* we expanded the Section 3 concerning the future development.

- When reading the paper I partially get very enthusiastic about this method, because it offers a nice new method for automatic monitoring. On the other hand, I keep on wondering what the added value of this method is over a traditional water level logger with millimetre accuracy, at more or less the same price. Such sensors are very robust, don't need frames, and additional constructions, have a very long battery life (weeks, months), and don't need any further processing.

*Authors' reply.* We are glad that the reviewer appreciates the proposed approach. We explained in the comments to Reviewer 2 the added value of the stage-cam compared to traditional water lever loggers.

- What I also wonder is whether this approach may be expanded with detection and monitoring of (anthropogenic) debris, such as woody debris, plastic pollution, or otherwise (van Lieshout et al., 2020). Then there's a clear added value over more traditional sensing equipment.

*Authors' reply.* We agree with the reviewer that optical sensors can be particularly effective in monitoring a variety of processes.
*Authors' action.* We emphasized in the Introduction this opportunity mentioning the suggested reference.

**Conclusions**

- In the conclusions the authors sate that their method allows for "supervising the stream area and banks". This is not elaborated on in the paper, so I suggest to either omit this statement or actually provide some additional analyses to support this in the paper.

*Authors' action.* we omitted this statement.

**Data and code availability**

- The data availability is not in line with HESS policy: https://www.hydrology-and-earth-system-sciences.net/policies/data_policy.html. I would strongly suggest to make the data openly available through one a repository. And otherwise follow HESS' policy to include a statement on why there are not available ("if the data are not publicly accessible, a detailed explanation of why this is the case is required").

*Authors' action.* we modified the data availability Section following the HEES policy.

**2. Specific comments:**

- Line 18-21: Maybe omit some references, seems a bit much.

*Authors' action.* Thanks for the suggestion, we accordingly modified the text.

- Line 26-48: Useful summary of other techniques and drawbacks, but can maybe be written more concisely.

*Authors' action.* Thanks for the suggestion, we accordingly modified the text.

- Line 85: Although not "purely hydrological", van Lieshout et al. (2020) recently demonstrated the potential of using cameras and deep learning for automatic plastic monitoring in rivers. Quite some lessons learned and practical considerations may be relevant for this manuscript as well.

*Authors' action.* Thanks for the suggestion, we accordingly modified the text.

- Line 122: How is the ROI automatically trimmed around it?

*Authors' reply*. Once the ROI is manually selected in the first frame, it is applied on all available frames leveraging the fact that both the pole and camera locations do not change in time.

- Line 137: What moving average is used? E.g. how many datapoints? How does the length of the window influence the accuracy?
- Line 138: Is the 90% quantile based on the entire dataseries? Or a subset (e.g. without outliers)?

*Authors' reply*. In this technical note we just applied the simplest filter approach, applying a common moving average (window width set to three) and setting a reasonable quantile threshold (90%). Varying the filtering parameter, we can observe a limited variability in the final output. However, such processing step will be accurately investigated in future works.

**References**

van Lieshout, Colin, et al. "Automated river plastic monitoring using deep learning and cameras." Earth and space science 7.8 (2020): e2019EA000960.

---

## Author Comment (AC4)

Hydrol. Earth Syst. Sci. Discuss., author comment AC1
https://doi.org/10.5194/hess-2021-36-AC1, 2021

[Figure]

**Authors' reply to CC, RC1, and RC2**

Simone Noto et al.

Author comment on "Technical Note: Low cost stage-camera system for continuous water level monitoring in ephemeral streams" by Simone Noto et al., Hydrol. Earth Syst. Sci. Discuss., https://doi.org/10.5194/hess-2021-36-AC1, 2021

**Authors' reply to:**

- Community Comment, CC, by Kenneth Chapman;
- Referee Comment 1, RC1, by Nils Kaplan;
- Referee Comment 2, RC2, by Anonymous Referee;

is included as supplement file.

Please also note the supplement to this comment:
https://hess.copernicus.org/preprints/hess-2021-36/hess-2021-36-AC1-supplement.pdf

---

## Author Comment (AC6)

Hydrol. Earth Syst. Sci. Discuss., author comment AC2
https://doi.org/10.5194/hess-2021-36-AC2, 2021

[Figure]

**Authors' reply to RC3**

Simone Noto et al.

Author comment on "Technical Note: Low cost stage-camera system for continuous water level monitoring in ephemeral streams" by Simone Noto et al., Hydrol. Earth Syst. Sci. Discuss., https://doi.org/10.5194/hess-2021-36-AC2, 2021

**Authors' reply to:**

Referee Comment 3, RC3, by Anonymous Referee;

is included as supplement file.

Please also note the supplement to this comment:
https://hess.copernicus.org/preprints/hess-2021-36/hess-2021-36-AC2-supplement.pdf